# Distributed Stochastic Optimization via Adaptive SGD

**Ashok Cutkosky**
Stanford University, USA[*]
cutkosky@google.com

**Róbert Busa-Fekete**
Yahoo! Research, New York, USA
busafekete@oath.com

## Abstract

Stochastic convex optimization algorithms are the most popular way to train machine learning models on large-scale data. Scaling up the training process of these models is crucial, but the most popular algorithm, Stochastic Gradient Descent (SGD), is a serial method that is surprisingly hard to parallelize. In this paper, we propose an efficient distributed stochastic optimization method by combining adaptivity with variance reduction techniques. Our analysis yields a linear speedup in the number of machines, constant memory footprint, and only a logarithmic number of communication rounds. Critically, our approach is a black-box reduction that parallelizes any serial online learning algorithm, streamlining prior analysis and allowing us to leverage the significant progress that has been made in designing adaptive algorithms. In particular, we achieve optimal convergence rates without any prior knowledge of smoothness parameters, yielding a more robust algorithm that reduces the need for hyperparameter tuning. We implement our algorithm in the Spark distributed framework and exhibit dramatic performance gains on large-scale logistic regression problems.

## 1 Setup

We consider a fundamental problem in machine learning, stochastic convex optimization:

$$\min_{w \in W} F(w) := \mathop{\mathbb{E}}_{f \sim \mathcal{D}}[f(w)] \tag{1}$$

Here, $W$ is a convex subset of $\mathbb{R}^d$ and $\mathcal{D}$ is a distribution over $L$-smooth convex functions $W \to \mathbb{R}$. We do not have direct access to $F$, and the distribution $\mathcal{D}$ is unknown, but we do have the ability to generate i.i.d. samples $f \sim \mathcal{D}$ through some kind of stream or oracle. In practice, each function $f \sim \mathcal{D}$ corresponds to a new datapoint in some learning problem. Algorithms for this problem are widely applicable: for example, in logistic regression the goal is to optimize $F(w) = \mathbb{E}[f(w)] = \mathbb{E}[\log(1 + \exp(-yw^T x))]$ when the $(x, y)$ pairs are the (feature vector, label) pairs coming from a fixed data distribution. Given a budget of $N$ oracle calls (e.g. a dataset of size $N$), we wish to find a $\hat{w}$ such that $F(\hat{w}) - F(w_\star)$ (called the *suboptimality*) is as small as possible as fast as possible using as little memory as possible, where $w_\star \in \operatorname{argmin} F$.

The most popular algorithm for solving (1) is Stochastic Gradient Descent (SGD), which achieves statistically optimal $O(1/\sqrt{N})$ suboptimality in $O(N)$ time and constant memory. However, in modern large-scale machine learning problems the number of data points $N$ is often gigantic, and so even the linear time-complexity of SGD becomes onerous. We need a *parallel* algorithm that runs in only $O(N/m)$ time using $m$ machines. We address this problem in this paper, evaluating solutions on three metrics: time complexity, space complexity, and communication complexity. Time complexity is the total time taken to process the data points. Space complexity is the amount of space required per

---

[*]now at Google

machine. Note that in our streaming model, an algorithm that keeps only the most recently seen data point in memory is considered to run in constant memory. Communication complexity is measured in terms of the number of "rounds" of communication in which all the machines synchronize. In measuring these quantities we often suppress all constants other than those depending on $N$ and $m$ and all logarithmic factors.

In this paper we achieve the ideal parallelization complexity (up to a logarithmic factor) of $\tilde{O}(N/m)$ time, $O(1)$ space and $\tilde{O}(1)$ rounds of communication, so long as $m < \sqrt{N}$. Further, in contrast to much prior work, our algorithm is a *reduction* that enables us to generically parallelize any serial online learning algorithm that obtains a sufficiently adaptive convergence guarantee (e.g. [8, 14, 6]) in a black-box way. This significantly simplifies our analysis by decoupling the learning rates or other internal variables of the serial algorithm from the parallelization procedure. This technique allows our algorithm to adapt to an unknown smoothness parameter $L$ in the problem, resulting in optimal convergence guarantees without requiring tuning of learning rates. This is an important aspect of the algorithm: even prior analyses that meet the same time, space and communication costs [9, 12, 13] require the user to input the smoothness parameter to tune a learning rate. Incorrect values for this parameter can result in failure to converge, not just slower convergence. In contrast, our algorithm automatically adapts to the true value of $L$ with no tuning. Empirically, we find that the parallelized implementation of a serial algorithm matches the performance of the serial algorithm in terms of sample-complexity, while bestowing significant runtime savings.

## 2 Prior Work

One popular strategy for parallelized stochastic optimization is minibatch-SGD [7], in which one computes $m$ gradients at a fixed point in parallel and then averages these gradients to produce a single SGD step. When $m$ is not too large compared to the variance in $\mathcal{D}$, this procedure gives a linear speedup in theory and uses constant memory. Unfortunately, minibatch-SGD obtains a communication complexity that scales as $\sqrt{N}$ (or $N^{1/4}$ for accelerated variants). In modern problems when $N$ is extremely large, this overhead is prohibitively large. We achieve a communication complexity that is logarithmic in $N$, allowing our algorithm to be run as a near-constant number of map-reduce jobs even for very large $N$. We summarize the state of the art for some prior algorithms algorithms in Table 1.

Many prior approaches to reducing communication complexity can be broadly categorized into those that rely on Newton's method and those that rely on the variance-reduction techniques introduced in the SVRG algorithm [11]. Algorithms that use Newton's method typically make the assumption that $\mathcal{D}$ is a distribution over quadratic losses [20, 22, 16, 21], and leverage the fact that the expected Hessian is constant to compute a Newton step in parallel. Although quadratic losses are an excellent starting point, it is not clear how to generalize these approaches to arbitrary non-quadratic smooth losses such as encountered in logistic regression.

Alternative strategies stemming from SVRG work by alternating between a "batch phase" in which one computes a very accurate gradient estimate using a large batch of examples and an "SGD phase" in which one runs SGD, using the batch gradient to reduce the variance in the updates [9, 12, 18, 10]. Our approach also follows this overall strategy (see Section 3 for a more detailed discussion of this procedure). However, all prior algorithms in this category make use of carefully specified learning rates in the SGD phase, while our approach makes use of *any* adaptive serial optimization algorithm, even ones that do not resemble SGD at all, such as [6, 14]. This results in a streamlined analysis and a more general final algorithm. Not only do we recover prior results, we can leverage the adaptivity of our base algorithm to obtain better results on sparse losses and to avoid any dependencies on the smoothness parameter $L$, resulting in a much more robust procedure.

The rest of this paper is organized as follows. In Section 3 we provide a high-level overview of our strategy. In Section 4 we introduce some basic facts about the analysis of adaptive algorithms using online learning, in Section 5 we sketch our intuition for combining SVRG and the online learning analysis, and in Section 6 we describe and analyze our algorithm. In Section 7 we show that the convergence rate is statistically optimal and show that a parallelized implementation achieves the stated complexities. Finally in Section 9 we give some experimental results.

Table 1: Comparison of distributed optimization algorithms with a dataset of size $N$ and $m$ machines. Logarithmic factors and all constants not depending on $N$ or $m$ have been dropped.

| Method | Quadratic Loss | Space | Communication | Adapts to $L$ |
|---|---|---|---|---|
| Newton inspired [20, 22, 16, 21] | Needed | $N/m$ | 1 | No |
| accel. minibatch-SGD [5] | Not Needed | 1 | $N^{1/4}$ | No |
| prior SVRG-like [9, 12, 18, 10] | Not Needed | 1 | 1 | No |
| **This work** | Not Needed | 1 | 1 | Yes |

## 3 Overview of Approach

Our overall strategy for parallelizing a serial SGD algorithm is based upon the stochastic variance-reduced gradient (SVRG) algorithm [11]. SVRG is a technique for improving the sample complexity of SGD given access to a stream of i.i.d. samples $f \sim \mathcal{D}$ (as in our setting), as well as the ability to compute *exact* gradients $\nabla F(v)$ in a potentially expensive operation. The basic intuition is to use an exact gradient $\nabla F(v)$ at some "anchor point" $v \in W$ as a kind of "hint" for what the exact gradient is at nearby points $w$. Specifically, SVRG leverages the theorem that $\nabla f(w) - \nabla f(v) + \nabla F(v)$ is an unbiased estimate of $\nabla F(w)$ with variance approximately bounded by $L(F(v) - F(w_\star))$ (see (8) in [11]). Using this fact, the SVRG strategy is:

1. Choose an "anchor point" $v = w_0$.

2. Compute an exact gradient $\nabla F(v)$ (this is an expensive operation).

3. Perform $T$ SGD updates: $w_{t+1} = w_t - \eta(\nabla f(w_t) - \nabla f(v) + \nabla F(v))$ for $T$ i.i.d. samples $f \sim \mathcal{D}$ using the fixed anchor $v$.

4. Choose a new anchor point $v$ by averaging the $T$ SGD iterates, set $w_0 = v$ and repeat 2-4.

By reducing the suboptimality of the anchor point $v$, the variance in the gradients also decreases, producing a virtuous cycle in which optimization progress reduces noise, which allows faster optimization progress. This approach has two drawbacks that we will address. First, it requires computing the exact gradient $\nabla F(v)$, which is impossible in our stochastic optimization setup. Second, prior analyses require specific settings for $\eta$ that incorporate $L$ and fail to converge with incorrect settings, requiring the user to manually tune $\eta$ to obtain the desired performance. To deal with the first issue, we can approximate $\nabla F(v)$ by averaging gradients over a mini-batch, which allows us to approximate SVRG's variance-reduced gradient estimate, similar to [9, 12]. This requires us to keep track of the errors introduced by this approximation. To deal with the second issue, we incorporate analysis techniques from online learning which allow us replace the constant step-size SGD with any adaptive stochastic optimization algorithm in a black-box manner. This second step forms the core of our theoretical contribution, as it both simplifies analysis and allows us to adapt to $L$.

The overall roadmap for our analysis has five steps:

1. We model the errors introduced by approximating the anchor-point gradient $\nabla F(v)$ by a minibatch-average as a "bias", so that we think of our algorithm as operating on slightly biased but low-variance gradient estimates.

2. Focusing first only the bias aspect, we analyze the performance of online learning algorithms with biased gradient estimates and show that so long as the bias is sufficiently small, the algorithm will still converge quickly (Section 4).

3. Next focusing on the variance-reduction aspect, we show that any online learning algorithm which enjoys a sufficiently adaptive convergence guarantee produces a similar "virtuous cycle" as observed with constant-step-size SGD in the analysis of SVRG, resulting in fast convergence (sketched in Section 5, proved in Appendices C and D).

4. Combine the previous three steps to show that applying SVRG using these approximate variance-reduced gradients and an adaptive serial SGD algorithm achieves $O(L/\sqrt{N})$ suboptimality using only $O(\sqrt{N})$ serial SGD updates (Sections 6 and 7).

5. Observe that the batch processing in step 3 can be done in parallel, that this step consumes the vast majority of the computation, and that it only needs to be repeated logarithmically many times (Section 7).

## 4 Biased Online Learning

A popular way to analyze stochastic gradient descent and related algorithms is through online learning [19]. In this framework, an algorithm repeatedly outputs vectors $w_t$ for $t = 1, 2, \ldots$ in some convex space $W$, and receives gradients $g_t$ such that $\mathbb{E}[g_t] = \nabla F(w_t)$ for some convex objective function $F$.[2] Typically one attempts to bound the linearized *regret*:

$$R_T(w_\star) = \sum_{t=1}^{T} g_t \cdot (w_t - w_\star)$$

Where $w_\star = \operatorname{argmin} F$. We can apply online learning algorithms to stochastic optimization via online-to-batch conversion [3], which tells us that

$$\mathbb{E}[F(\overline{w}) - F(w_\star)] \leq \frac{\mathbb{E}[R_T(w_\star)]}{T}$$

where $\overline{w} = \frac{1}{T} \sum_{t=1}^{T} w_t$.

Thus, an algorithm that guarantees small regret immediately guarantees convergence in stochastic optimization. Online learning algorithms typically obtain some sort of (deterministic!) guarantee like

$$R_T(w_\star) \leq R(w_\star, \|g_1\|, \ldots, \|g_T\|)$$

where $R$ is increasing in each $\|g_t\|$. For example, when the convex space $W$ has diameter $D$, AdaGrad [8] obtains $R_T(w_\star) \leq D\sqrt{2 \sum_{t=1}^{T} \|g_t\|^2}$.

As foreshadowed in Section 3, we will need to consider the case of *biased gradients*. That is, $\mathbb{E}[g_t] = \nabla F(w_t) + b_t$ for some unknown bias vector $b_t$. Given these biased gradients, a natural question is: to what extent does controlling the regret $R_T(w_\star) = \sum_{t=1}^{T} g_t \cdot (w_t - w_\star)$ affect our ability to control the suboptimality $\mathbb{E}[\sum_{t=1}^{T} F(w_t) - F(w_\star)]$? We answer this question with the following simple result:

**Proposition 1.** *Define* $R_T(w_\star) = \sum_{t=1}^{T} g_t \cdot (w_t - w_\star)$ *and* $\overline{w} = \frac{1}{T} \sum_{t=1}^{T} w_t$ *where* $\mathbb{E}[g_t] = \nabla F(w_t) + b_t$. *Then*

$$\mathbb{E}[F(\overline{w}) - F(w_\star)] \leq \frac{\mathbb{E}[R_T(w_\star)]}{T} + \frac{1}{T} \sum_{t=1}^{T} \mathbb{E}[\|b_t\|(\|w_t - w_\star\|)]$$

*If the domain $V$ has diameter $D$, then* $\mathbb{E}[F(\overline{w}) - F(w_\star)] \leq \frac{\mathbb{E}[R_T(w_\star)]}{T} + \frac{D}{T} \sum_{t=1}^{T} \mathbb{E}[\|b_t\|]$

Our main convergence results will require algorithms with regret bounds of the form $R(w_\star) \leq \psi(w_\star)\sqrt{\sum_{t=1}^{T} \|g_t\|^2}$ or $R(w_\star) \leq \psi(w_\star)\sqrt{\sum_{t=1}^{T} \|g_t\|}$ for various $\psi$. This is an acceptable restriction because there are many examples of such algorithms, including AdaGrad [8], SOLO [15], PiSTOL [14] and FreeRex [6]. Further, in Proposition 3 we show a simple trick to remove the dependence on $\|w_t - w_\star\|$, allowing our results to extend to unbounded domains.

## 5 Variance-Reduced Online Learning

In this section we sketch an argument that using variance reduction in conjunction with a online learning algorithm guaranteeing regret $R(w_\star) \leq \psi(w_\star)\sqrt{\sum_{t=1}^{T} \|g_t\|^2}$ results in a very fast convergence of $\sum_{t=1}^{T} \mathbb{E}[F(w_t) - F(w_\star)] = O(1)$ up to log factors. A similar result holds for regret guarantees like $R(w_\star) \leq \psi(w_\star)\sqrt{\sum_{t=1}^{T} \|g_t\|}$ via a similar argument, which we leave to Appendix D. To do

this we make use of a critical lemma of variance reduction which asserts that a variance-reduced gradient estimate $g_t$ of $\nabla F(w_t)$ with anchor point $v_t$ has $\mathbb{E}[\|g_t\|^2] \leq L(F(w_t) + F(v_t) - 2F(w_\star))$ up to constants. This gives us the following informal result:

**Proposition 2.** *[Informal statement of Proposition 8] Given a point $w_t \in W$, let $g_t$ be an unbiased estimate of $\nabla F(w_t)$ such that $\mathbb{E}[\|g_t\|^2] \leq L(F(w_t) + F(v_t) - 2F(w_\star))$. Suppose $w_1, \ldots, w_T$ are generated by an online learning algorithm with regret at most $R(w_\star) \leq \psi(w_\star)\sqrt{\sum_{t=1}^{T} \|g_t\|^2}$. Then*

$$\mathbb{E}\left[\sum_{t=1}^{T} F(w_t) - F(w_\star)\right] = O\left(L\psi(w_\star)^2 + \psi(w_\star)\sqrt{\sum_{t=1}^{T} L\,\mathbb{E}[F(v_t) - F(w_\star)]}\right) \qquad (2)$$

*Proof.* The proof is remarkably simple, and we sketch it in one line here. The full statement and proof can be found in Appendix D.

$$\mathbb{E}\left[\sum_{t=1}^{T} F(w_t) - F(w_\star)\right] \leq \psi(w_\star)\,\mathbb{E}\left[\sqrt{\sum_{t=1}^{T} \|g_t\|^2}\right]$$

$$\leq \psi(w_\star)\sqrt{\sum_{t=1}^{T} L\,\mathbb{E}[F(w_t) - F(w_\star) + F(v_t) - F(w_\star)]}$$

Now square both sides and use the quadratic formula to solve for $\mathbb{E}\left[\sum_{t=1}^{T} F(w_t) - F(w_\star)\right]$. $\qquad \square$

Notice that in Proposition 2, the online learning algorithm's regret guarantee $\psi(w_\star)\sqrt{\sum_{t=1}^{T} \|g_t\|^2}$ does not involve the smoothness parameter $L$, and yet nevertheless $L$ shows up in equation (2). It is this property that will allow us to adapt to $L$ without requiring any user-supplied information.

---

**Algorithm 1** SVRG OL (**SVRG** with **O**nline **L**earning)

---

1: **Initialize:** Online learning algorithm $\mathcal{A}$; Batch size $\hat{N}$; epoch lengths $0 = T_0, \ldots, T_K$; Set $T_{a:b} = \sum_{i=a}^{b} T_i$.
2: Get initial vector $w_1$ from $\mathcal{A}$, set $v_t \leftarrow w_1$.
3: **for** $k = 1$ **to** $K$ **do**
4: $\quad$ Sample $\hat{N}$ functions $f_1, \ldots, f_{\hat{N}} \sim \mathcal{D}$
5: $\quad$ $\nabla \hat{F}(v_k) \leftarrow \frac{1}{\hat{N}} \sum_{i=1}^{\hat{N}} \nabla f_i(v_k)$
6: $\quad$ (this step can be done in parallel).
7: $\quad$ **for** $t = T_{0:k-1} + 1$ **to** $T_{0:k}$ **do**
8: $\quad\quad$ Sample $f \sim \mathcal{D}$.
9: $\quad\quad$ Give $g_t = \nabla f(w_t) - \nabla f(v_k) + \nabla \hat{F}(v_k)$ to $\mathcal{A}$.
10: $\quad\quad$ Get updated vector $w_{t+1}$ from $\mathcal{A}$.
11: $\quad$ **end for**
12: $\quad$ $v_{k+1} \leftarrow \frac{1}{T_k} \sum_{t=T_{0:k-1}+1}^{T_{0:k}} w_t$.
13: **end for**

---

Variance reduction allows us to generate estimates $g_t$ satisfying the hypothesis of Proposition 2, so that we can control our convergence rate by picking appropriate $v_t$s. We want to change $v_t$ very few times because changing anchor points requires us to compute a high-accuracy estimate of $\nabla F(v_t)$. Thus we change $v_t$ only when $t$ is a power of 2 and set $v_{2^n}$ to be the average of the last $2^{n-1}$ iterates $w_t$. By Jensen, this allows us to bound $\sum_{t=1}^{T} \mathbb{E}[F(v_t) - F(w_\star)]$ by $\sum_{t=1}^{T} \mathbb{E}[F(w_t) - F(w_\star)]$, and so applying Proposition 2 we can conclude $\sum_{t=1}^{T} \mathbb{E}[F(w_t) - F(w_\star)] = O(1)$.

# 6 SVRG with Online Learning

With the machinery of the previous sections, we are now in a position to derive and analyze our main algorithm, presented in SVRG OL.

SVRG OL implements the procedure described in Section 3. For each of a series of $K$ rounds, we compute a batch gradient estimate $\nabla \hat{F}(v_k)$ for some "anchor point" $v_k$. Then we run $T_k$ iterations of an online learning algorithm. To compute the $t^{th}$ gradient $g_t$ given to the online learning algorithm in response to an output point $w_t$, SVRG OL approximates the variance-reduction trick of SVRG, setting $g_t = \nabla f(w_t) - \nabla f(v_k) + \nabla \hat{F}(v_k)$ for some new sample $f \sim \mathcal{D}$. After the $T_k$ iterations have elapsed, a new anchor point $v_{k+1}$ is chosen and the process repeats.

In this section we characterize SVRG OL's performance when the base algorithm $\mathcal{A}$ has a regret guarantee of $\psi(w_\star)\sqrt{\sum_{t=1}^T \|g_t\|^2}$. We can also perform essentially similar analysis for regret guarantees like $\psi(w_\star)\sqrt{\sum_{t=1}^T \|g_t\|}$, but we postpone this to Appendix E.

In order to analyze SVRG OL, we need to bound the error $\|\nabla \hat{F}(v_k) - \nabla F(v_k)\|$ uniformly for all $k \leq K$. This can be accomplished through an application of Hoeffding's inequality:

**Lemma 1.** *Suppose that $\mathcal{D}$ is a distribution over $G$-Lipschitz functions. Then with probability at least $1 - \delta$, $\max_k \|\nabla \hat{F}(v_k) - \nabla F(v_k)\| \leq \sqrt{\frac{2G^2 \log(K/\delta) + G^2}{\hat{N}}}$.*

The proof of Lemma 1 is deferred to Appendix A. The following Theorem is now an immediate consequences of the concentration bound Lemma 1 and Propositions 8 and 9 (see Appendix).

**Theorem 1.** *Suppose the online learning algorithm $\mathcal{A}$ guarantees regret $R_T(w_\star) \leq \psi(w_\star)\sqrt{\sum_{t=1}^T \|g_t\|^2}$. Set $b_t = \|\nabla \hat{F}(v_k) - \nabla F(v_k)\|$ for $t \in [T_{0:k-1} + 1, T_{1:k}]$ (where $T_{a:b} := \sum_{i=a}^b T_i$). Suppose that $T_k/T_{k-1} \leq \rho$ for all $k$. Then for $\overline{w} = \frac{1}{T}\sum_{t=1}^T w_t$,*

$$\mathbb{E}[F(\overline{w}) - F(w_\star)] \leq \frac{32(1+\rho)\psi(w_\star)^2 L}{T} + \frac{2\sum_{t=1}^T \mathbb{E}[\|b_t\|(\|w_t - w_\star\|)]}{T}$$
$$+ \frac{2\psi(w_\star)\sqrt{8LT_1 \mathbb{E}[F(v_1) - F(w_\star)] + 2\sum_{t=1}^T \mathbb{E}[\|b_t\|^2]}}{T}$$

*In particular, if $\mathcal{D}$ is a distribution over $G$-Lipschitz functions, then with probability at least $1 - \frac{1}{T}$ we have $\|b_t\| \leq \sqrt{\frac{2G^2 \log(KT) + G^2}{\hat{N}}}$ for all $t$. Further, if $\hat{N} > T^2$ and $V$ has diameter $D$, then this implies*

$$\mathbb{E}[F(\overline{w}) - F(w_\star)] \leq \frac{32(1+\rho)\psi(w_\star)^2 L}{T} + \frac{4\psi(w_\star)\sigma\sqrt{\log(KT)}}{T\sqrt{T}}$$
$$+ \frac{8\psi(w_\star)\sqrt{LT_1 \mathbb{E}[F(v_1) - F(w_\star)]}}{T} + \frac{GD}{T} + 2\frac{G\sqrt{2\log(KT) + 1}D}{T}$$
$$= O\left(\frac{\sqrt{\log(KT)}}{T}\right)$$

We note that although this theorem requires a finite diameter for the second result, we present a simple technique to deal with unbounded domains and retain the same result in Appendix D

## 7 Statistical and Computational Complexity

In this section we describe how to choose the batch size $\hat{N}$ and epoch sizes $T_k$ in order to obtain optimal statistical complexity and computational complexity. The total amount of data used by SVRG OL is $N = K\hat{N} + T_{0:K} = K\hat{N} + T$. If we choose $\hat{N} = T^2$, this is $O(K\hat{N})$. Set $T_k = 2T_{k-1}$, with some $T_1 > 0$ so that $\rho = \max T_k/T_{k-1} = 2$ and $O(K = \log(N))$. Then our Theorem 1 guarantees suboptimality $O(\sqrt{\log(TK)}/T)$, which is $O(\sqrt{K\log(TK)}/\sqrt{K\hat{N}}) = O(\sqrt{K\log(TK)}/\sqrt{N})$. This matches the optimal $O(1/\sqrt{N})$ up to logarithmic factors and constants.

The parallelization step is simple: we parallelize the computation of $\nabla \hat{F}(v_k)$ by having $m$ machines compute and sum gradients for $\hat{N}/m$ new examples each, and then averaging these $m$ sums together

on one machine. Notice that this can be done with constant memory footprint by streaming the examples in - the algorithm will not make any further use of these examples so it's safe to forget them. Then we run the $T_k$ steps of the inner loop in serial, which again can be done in constant memory footprint. This results in a total runtime of $O(K\hat{N}/m + T)$ - a linear speedup so long as $m < KN/T$. For algorithms with regret bounds matching the conditions of Theorem 1, we get optimal convergence rates by setting $\hat{N} = T^2$, in which case our total data usage is $N = O(K\hat{N})$. This yields the following calculation:

**Theorem 2.** *Set $T_k = 2T_{k-1}$. Suppose the base optimizer $\mathcal{A}$ in SVRG OL guarantees regret $R_T(w_\star) \le \psi(w_\star)\sqrt{\sum_{t=1}^T \|g_t\|^2}$, and the domain $W$ has finite diameter $D$. Let $\hat{N} = \Theta(T^2)$ and $N = K\hat{N} + T$ be the total number of data points observed. Suppose we compute the batch gradients $\nabla \hat{F}(v_k)$ in parallel on $m$ machines with $m < \sqrt{N}$. Then for $\overline{w} = \frac{1}{T}\sum_{t=1}^T w_t$ we obtain*

$$\mathbb{E}[F(\overline{w}) - F(w_\star)] = \tilde{O}\left(\frac{1}{\sqrt{N}}\right)$$

*in time $\tilde{O}(N/m)$, and space $O(1)$, and $K = \tilde{O}(1)$ communication rounds.*

## 8 Implementation

### 8.1 Linear Losses and Dense Batch Gradients

Many losses of practical interest take the form $f(w) = \ell(w \cdot x, y)$ for some label $y$ and feature vector $x \in \mathbb{R}^d$ where $d$ is extremely large, but $x$ is $s$-sparse. These losses have the property that $\nabla f(w) = \ell'(w \cdot x, y)x$ is also $s$-sparse. Since $d$ can often be very large, it is extremely desirable to perform all parameter updates in $O(s)$ time rather than $O(d)$ time. This is relatively easy to accomplish for most SGD algorithms, but our strategy involves correcting the variance in $\nabla f(w)$ using a *dense* batch gradient $\nabla \hat{F}(v_k)$ and so we are in danger of losing the significant computational speedup that comes from taking advantage of sparsity. We address this problem through an importance-sampling scheme.

Suppose the $i$th coordinate of $x$ is non-zero with probability $p_i$. Given a vector $v$, let $I(v)$ be the vector whose $i$th component is 0 if $w_i = 0$, or $1/p_i$ is $w_i \ne 0$. Then $\mathbb{E}[I(\nabla f(w))]$ is equal to the all-ones vector. Using this notation, we replace the variance-reduced gradient estimate $\nabla f(w) - \nabla f(v_k) + \nabla \hat{F}(v_k)$ with $\nabla f(w) - \nabla f(v_k) + \nabla \hat{F}(v_k) \odot I(\nabla f(w))$, where $\odot$ indicates component-wise multiplication. Since $\mathbb{E}[I(\nabla f(w))]$ is the all-ones vector, $\mathbb{E}[\nabla \hat{F}(v_k) \odot I(\nabla f(w))] = \nabla \hat{F}(v_k)$ and so the expected value of this estimate has not changed. However, it is clear that the *sparsity* of the estimate is now equal to the sparsity of $\nabla f(w)$. Performing this transformation introduces additional variance into the estimate, and could slow down our convergence by a constant factor. However, we find that even with this extra variance we still see impressive speedups (see Section 9).

### 8.2 Spark implementation

Implementing our algorithm in the Spark environment is fairly straightforward. SVRG OL switches between two phases: a batch gradient computation phase and a serial phase in whicn we run the online learning algorithm. The serial phase is carried out by the driver while the batch gradient is computed by executors. We initially divide the training data into $C$ approximately 100M chunks, and we use $\min(1000, C)$ executors. Tree aggregation with depth of 5 is used to gather the gradient from the executors, which is similar to the operation implemented by Vowpal Wabbing (VW) [1]. We use asynchronous collects to move the instances used in the next serial SGD phase of SVRG OL to the driver while the batch gradient is being computed. We used feature hashing with 23 bits to limit memory consumption.

### 8.3 Batch sizes

Our theoretical analysis asks for exponentially increasing serial phase lengths $T_k$ and a batch size of of $\hat{N} = T^2$. In practice we use slightly different settings. We have a constant serial phase length $T_k = T_0$ for all $k$, and an increasing batch size $\hat{N}_k = kC$ for some constant $C$. We usually set

$C = T_0$. The constant $T_k$ is motivated by observing that the requirement for exponentially increasing $T_k$ comes from a desire to offset potential poor performance in the first serial phase (which gives the dependence on $T_1$ in Theorem 1). In practice we do not expect this to be an issue. The increasing batch size is motivated by the empirical observation that earlier serial phases (when we are farther from the optimum) typically do not require as accurate a batch gradient in order to make fast progress.

Table 2: Statistics of the datasets. The compressed size of the data is reported. B=Billion, M=Million

| Data | # Instance | | Data size (Gb) | | # Features | Avg # feat | % positives |
|---|---|---|---|---|---|---|---|
| – | Train | Test | Train | Test | | | |
| KDD10 | 19.2M | 0.74M | 0.5 | 0.02 | 29 890 095 | 29.34 | 86.06% |
| KDD12 | 119.7M | 29.9M | 1.6 | 0.5 | 54 686 452 | 11.0 | 4.44% |
| ADS SMALL | 1.216B | 0.356B | 155.0 | 40.3 | 2 970 211 | 92.96 | 8.55% |
| ADS LARGE | 5.613B | 1.097B | 1049.1 | 486.1 | 12 133 899 | 95.72 | 9.42% |
| EMAIL | 1.236B | 0.994B | 637.4 | 57.6 | 37 091 273 | 132.12 | 18.74% |

## 9 Experiments

To verify our theoretical results, we carried out experiments on large-scale (order 100 million datapoints) public datasets, such as KDD10 and KDD12 [3] and on proprietary data (order 1 billion datapoints), such as click-prediction for ads [4] and email click-prediction datasets [2]. The main statistics of the datasets are shown in Table 2. All of these are large datasets with sparse features, and heavily imbalanced in terms of class distribution. We solved these binary classification tasks with logistic regression. We tested two well-know scalable logistic regression implementation: Spark ML 2.2.0 and Vowpal Wabbit 7.10.0 (VW) [4]. To optimize the logistic loss we used the L-BFGS algorithm implemented by both packages. We also tested minibatch SGD and non-adaptive SVRG implementations. However, we observe that the relationship between non-adaptive SVRG updates and the updates in our algorithm are analogous to the relationship between the updates in constant-step-size SGD and (for example) AdaGrad. Since our experiments involved sparse high-dimensional data, adaptive step sizes are very important and one should not expect these algorithms to be competitive (and indeed they were not).

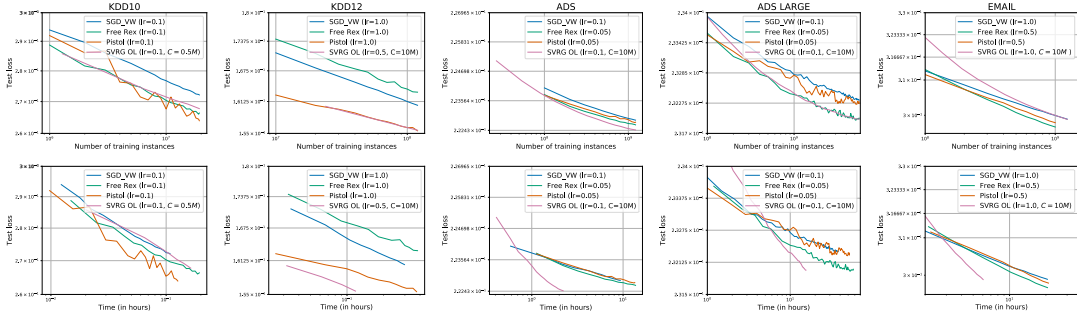

Figure 1: Test loss of three SGD algorithms (PiSTOL [14], Vowpal Wabbit (labeled as SGD VW) [17] and FreeRex [6]) and SVRG OL (labeled as SVRG OL, using FreeRex as the base optimizer) on the benchmark datasets.

First we compared SVRG OL to several non-parallelized baseline SGD optimizers on the different datasets. We plot the loss a function of the number of datapoints processed, as well as the total runtime (Figure 1). Measuring the number of datapoints processed gives us a sense of the statistical efficiency of the algorithm and gives a metric that is independent of implementation quality details. We see that, remarkably, SVRG OL's actually performs well as a function of number of datapoints

processed and so is a competitive *serial* algorithm before even any parallelization. Thus it is no surprise that we see significant speedups when the batch computation is parallelized.

To assess the trend of the speed-up with the size of the training data, we plotted the relative speed-up of SVRG OL versus FreeRex which is used as base optimizer in SVRG OL. Figure 2 shows the fraction of running time of non-parallel and parallel algorithms needed to achieve the same performance in terms of test loss. The x-axis scales with the running time of the parallel SVRG OL algorithm. The speed-up increases with training time, and thus the number of training instances processed. This result suggests that our method will indeed match with the theoretical guarantees in case of large enough datasets, although this trend is hard to verify rigorously in our test regime.

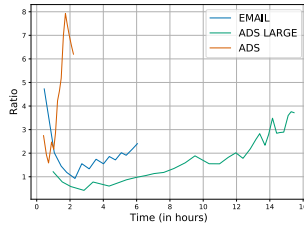

Figure 2: The speed-up ratio of SVRG OL versus FreeRex on various datasets.

In our second experiment, we proceed to compare SVRG OL to Spark ML and VW in Table 4. These two LBFGS-based algorithms were superior in all metrics to minibatch SGD and non-adaptive SVRG algorithms and so we report only the comparison to Spark ML and VW (see Section F for full results). We measure the number of communication rounds, the total training error, the error on a held-out test set, the Area Under the Curve (AUC), and total runtime in minutes. Table 4 illustrates that SVRG OL compares well to both Spark ML and VW. Notably, SVRG OL uses dramatically fewer communication rounds. On the smaller KDD datasets, we also see much faster runtimes, possibly due to overhead costs associated with the other algorithms. It is important to note that our SVRG OL makes only one pass over the dataset, while the competition makes one pass per communication round, resulting in 100s of passes. Nevertheless, we obtain competitive final error due to SVRG OL's high statistical efficiency.

Table 3: Average loss and AUC achieved by Logistic regression implemented in Spark ML, VW and SVRG OL. "Com." refers to number of communication rounds and time is measured in minutes. The results on KDD10, ADS LARGE and EMAIL data is presented in App. F due to lack of space.

| Dataset | KDD12 | | | | | ADS SMALL | | | | |
|---|---|---|---|---|---|---|---|---|---|---|
| | Com. | Train | Test | AUC | Time | Com. | Train | Test | AUC | Time |
| Spark ML | 100 | 0.15756 | 0.15589 | 75.485 | 36 | 100 | 0.23372 | 0.22288 | 83.356 | 42 |
| Spark ML | 550 | 0.15755 | 0.15570 | 75.453 | 180 | 500 | 0.23365 | 0.22286 | 83.365 | 245 |
| VW | 100 | 0.15398 | 0.15725 | 77.871 | 44 | 100 | 0.23381 | 0.22347 | 83.214 | 114 |
| VW | 500 | **0.14866** | 0.15550 | **78.881** | 150 | 500 | 0.23157 | 0.22251 | **83.499** | 396 |
| SVRG OL | 4 | 0.152740 | **0.154985** | 78.431 | 8 | 14 | **0.23147** | **0.22244** | 83.479 | 94 |

## 10   Conclusion

We have presented SVRG OL, a generic stochastic optimization framework which combines adaptive online learning algorithms with variance reduction to obtain communication efficiency in parallel architectures. Our analysis significantly streamlines previous work by making black-box use of any adaptive online learning algorithm, thus disentangling the variance-reduction and serial phases of SVRG algorithms. We require only a logarithmic number of communication rounds, and we automatically adapt to an unknown smoothness parameter, yielding both fast performance and robustness to hyperparameter tuning. We developed a Spark implementation of SVRG OL and solved real large scale sparse learning problems with competitive performance to L-BFGS implemented by VW and Spark ML.

## Footnotes

[2]The online learning literature often allows for adversarially generated $g_t$, but we consider only stochastically generated $g_t$ here.

[3] https://www.csie.ntu.edu.tw/~cjlin/libsvmtools/datasets/binary.html

[4] https://github.com/JohnLangford/vowpal_wabbit/releases/tag/7.10

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
