[Supplementary Material]

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

# Supplementary material for "Distributed Stochastic Optimization via Adaptive SGD"

The appendix starts with the proof of Lemma 1 in Appendix A. Next, in Section B we provide the proofs from Section 4. In Section C we review prior results about the properties of smooth convex functions as variance reduction. In Section D we then combine the previous two sections' results to prove the convergence of our SVRG OL. Finally, in Section F we provide additional information about our experiments, including statistics of the various datasets as well as a complete reporting of the performance of all tested algorithms.

## A    Proof of Lemma 1

*Proof.* The assumption on $\mathcal{D}$ implies that $\|\nabla f(v_k)\|$ is bounded by $G$ and so $\nabla f(v_k)$ is $G^2$-subgaussian. Therefore we can apply the Hoeffding and union bounds to obtain tail bounds on $\nabla f(v_k)$:

$$\mathrm{Prob}\left(\left\|\frac{1}{\hat{N}}\sum_{i=1}^{\hat{N}}\nabla f_i(v_k) - \mathbb{E}[\nabla f(v_k)]\right\| \geq \epsilon \text{ for all } k\right)$$

$$\leq K\exp\left[-\left(\hat{N}\epsilon - G\sqrt{\hat{N}}\right)^2/(2\hat{N}G^2)\right]$$

rearranging, with probability at least $1 - \delta$, for all $k$ we have

$$\left\|\frac{1}{\hat{N}}\sum_{i=1}^{N}\nabla f_i(v_k) - \mathbb{E}[\nabla f(v_k)]\right\| \leq \sqrt{\frac{2G^2\log(K/\delta) + G^2}{\hat{N}}}$$

as desired. $\qquad\square$

## B    Proofs from Section 4

**Proposition 1.** *Define $R_T(w_\star) = \sum_{t=1}^{T} g_t \cdot (w_t - w_\star)$ and $\overline{w} = \frac{1}{T}\sum_{t=1}^{T} w_t$ where $\mathbb{E}[g_t] = \nabla F(w_t) + b_t$. Then*

$$\mathbb{E}[F(\overline{w}) - F(w_\star)] \leq \frac{\mathbb{E}[R_T(w_\star)]}{T} + \frac{1}{T}\sum_{t=1}^{T}\mathbb{E}[\|b_t\|(\|w_t - w_\star\|)]$$

*If the domain $V$ has diameter $D$, then $\mathbb{E}[F(\overline{w}) - F(w_\star)] \leq \frac{\mathbb{E}[R_T(w_\star)]}{T} + \frac{D}{T}\sum_{t=1}^{T}\mathbb{E}[\|b_t\|]$*

*Proof.* The proof follows from Cauchy-Schwarz, triangle inequality, and convexity of $F$:

$$\mathbb{E}[R_T] = \mathbb{E}\left[\sum_{t=1}^{T} g_t(w_t - w_\star)\right]$$

$$= \mathbb{E}\left[\sum_{t=1}^{T}\nabla F(w_t)\cdot(w_t - w_\star) + b_t\cdot(w_t - w_\star)\right]$$

$$\geq \mathbb{E}\left[\sum_{t=1}^{T}F(w_t) - F(w_\star) - \|b_t\|\|w_t - w_\star\|\right]$$

Now rearrange and apply Jensen's inequality to recover the first line of the Proposition. The second statement follows from observing that $\|w_t - w_\star\| \leq D$. $\qquad\square$

Proposition 1 shows that the suboptimality increases when both $\|b_t\|$ and $\|w_t\|$ becomes large. Although the online learning algorithm does not have the ability to control $\|b_t\|$, it does have the ability to control $\|w_t\|$, and so we can design a reduction to compensate for $\|b_t\|$. The reduction is simple: instead of $g_t$, provide the algorithm with $g_t + B\frac{w_t}{\|w_t\|}$, where $B$ is a bound such that $B \geq \|b_t\|$ for all $t$, and by abuse of notation we take $w_t/\|w_t\| = 0$ when $w_t = 0$. Proposition 3 below, tells us that, so long as we know the bound $B$, we can obtain an increase in suboptimality that depends only on $B$ and not $w_t$.

**Proposition 3.** *Suppose an online learning algorithm $\mathcal{A}$ guarantees regret $R_T(w_\star) \leq R(w_\star, \|g_1\|, \ldots, \|g_T\|)$, where $R$ is an increasing function of each $\|g_t\|$. Then if we run $\mathcal{A}$ on gradients $g_t + B\frac{w_t}{\|w_t\|}$, we obtain:*

$$\mathbb{E}\left[\sum_{t=1}^{T}F(w_t) - F(w_\star)\right] \leq R(w_\star, \|g_1\| + B, \ldots, \|g_T\| + B) + 2TB$$

*and, thus $\mathbb{E}[F(\overline{w}) - F(w_\star)] \leq \frac{1}{T}R(w_\star, \|g_1\| + B, \ldots, \|g_T\| + B) + 2B$.*

*Proof.* Observe that $B\frac{w_t}{\|w_t\|} = \nabla h(w_t)$ where $h(x) = B\|x\|$, so that by convexity we have

$$\mathbb{E}\left[\sum_{t=1}^{T} F(w_t) - F(w_\star) + b_t(w_t - w_\star) + B\|w_t\| - B\|w_\star\|\right]$$

$$\leq R\left(w_\star, \left\|g_1 + B\frac{w_t}{\|w_t\|}\right\|, \cdots\right)$$

$$\leq R(w_\star, \|g_1\| + B, \ldots, \|g_T\| + B)$$

Now observe that $b_t(w_t - w_\star) + B\|w_t\| - B\|w_\star\| \geq -2B\|w_\star\|$ to obtain:

$$\mathbb{E}\left[\sum_{t=1}^{T} F(w_t) - F(w_\star)\right] - 2TB\|w_\star\|$$

$$\leq R(w_\star, \|g_1\| + B, \ldots, \|g_T\| + B)$$

$$\mathbb{E}\left[\sum_{t=1}^{T} F(w_t) - F(w_\star)\right]$$

$$\leq R(w_\star, \|g_1\| + B, \ldots, \|g_T\| + B) + 2TB$$

Finally, use Jensen's inequality to concude the Proposition $\qquad\square$

## C   Smooth Losses

In the following sections we consider applying an online learning algorithm to gradients $g_t$ of the form $g_t = \nabla f_t(w_t) - \nabla f_t(\hat{x}) + \nabla F(\hat{x}) + b_t$ where each $f_t$ is an i.i.d. smooth convex function with $\mathbb{E}[f_t] = F$ and $\hat{x}$ is some fixed vector. In order to leverage the structure of this kind of $g_t$, we'll need two lemmas from the literature about smooth convex functions (which can be found, for example, in [11]):

**Lemma 2.** *If $f$ is an $L$-smooth convex function and $x, y$ are fixed vectors, then*

$$\|\nabla f(x) - \nabla f(y)\|^2 \leq 2L(f(x) - f(y) - \nabla f(y)(x - y))$$

*Proof.* Set $\tilde{f}(w) = f(w) - f(y) - \nabla f(y) \cdot (w - y)$. Then $\tilde{f}$ is still convex and $L$-smooth and $\nabla \tilde{f}(y) = 0$ so that $\tilde{f}(y) \leq \tilde{f}(q)$ for all $q$. Therefore We have

$$0 = \tilde{f}(y) \leq \inf_z \tilde{f}(x - z\nabla\tilde{f}(x))$$

$$\leq \inf_z \tilde{f}(x) - z\|\nabla\tilde{f}(x)\|^2 + \frac{L}{2}z^2\|\nabla\tilde{f}(x)\|^2$$

$$= \tilde{f}(x) - \frac{1}{2L}\|\nabla\tilde{f}(x)\|^2$$

$$= f(x) - f(y) - \nabla f(y) \cdot (x - y) - \frac{1}{2L}\|\nabla f(x) - \nabla f(y)\|^2$$

from which the Lemma follows. $\qquad\square$

We can use Lemma 2 to show the following useful fact:

**Lemma 3.** *Suppose $D$ is a distribution over $L$-smooth convex functions $f$ and $F = \mathbb{E}[f]$. Let $w_\star \in \operatorname{argmin} F$. Then for all $x$ we have $\mathbb{E}[\|\nabla f(x) - \nabla f(w_\star)\|^2] \leq 2L[F(x) - F(w_\star)]$.*

*Proof.* From Lemma 2 we have

$$\mathbb{E}[\|\nabla f(x) - \nabla f(w_\star)\|^2] \leq 2L\,\mathbb{E}[f(x) - f(w_\star) - \nabla f(w_\star) \cdot (x - w_\star)]$$

and now the result follows since $E[\nabla f(w_\star)] = \nabla f(w_\star) = 0$. $\qquad\square$

With these in hand, we can prove

**Proposition 4.** *With $g_t = \nabla f(w_t) - \nabla f(v_k) + \nabla F(v_k) + b_t$ for some points $w_t, v_k \in W$ and $b_t \in \mathbb{R}$, we have*

$$\mathbb{E}[\|g_t\|^2] \leq 8L\,\mathbb{E}[F(w_t) - F(w_\star)] + 8L\,\mathbb{E}[F(v_k) - F(w_\star)] + 2\,\mathbb{E}[\|b_t\|^2]$$

*Proof.* Observe that $\mathbb{E}[\nabla f(w_\star) - \nabla f(v_k)] = \nabla F(w_\star) - \nabla F(v_k)$, so that by Lemma 3

$$\mathbb{E}[\|\nabla f(w_\star) - \nabla f(v_k) + \nabla F(v_k) - \nabla F(w_\star)\|^2 \le \mathbb{E}[\|\nabla f(w_\star) - \nabla f_t(v_k)\|^2] \\ \le 2L(F(v_k) - F(w_\star))$$

Using this we have

$$\begin{aligned}
\mathbb{E}[\|g_t\|^2] &\le \mathbb{E}[\|\nabla f(w_t) - \nabla f(v_k) + \nabla F(v_k) + b_t\|^2] \\
&\le \mathbb{E}[2\|\nabla f(w_t) - \nabla f(v_k) + \nabla F(v_k)\|^2 + 2\|b_t\|^2] \\
&\le \mathbb{E}[4\|\nabla f(w_t) - \nabla f(w_\star)\|^2 + 4\|\nabla f(w_\star) - \nabla f(v_k) + \nabla F(v_k) - \nabla F(w_\star)\|^2 + 2\|b_t\|^2] \\
&\le 8L\,\mathbb{E}[F(w_t) - F(w_\star)] + 4\,\mathbb{E}[\|\nabla f(w_\star) - \nabla f(v_k)\|^2] + 2\,\mathbb{E}[\|b_t\|^2] \\
&\le 8L\,\mathbb{E}[F(w_t) - F(w_\star)] + 8L\,\mathbb{E}[F(v_k) - F(w_\star)] + 2\,\mathbb{E}[\|b_t\|^2]
\end{aligned}$$

$\square$

# D  Biased Online Learning with SVRG updates

In this section are finally prepared to analyze SVRG OL. In order to do so, we restate the algorithm as Algorithm 2. This algorithm is identical to SVRG OL, but we have introduced the additional notation $b_t = \nabla\hat{F}(v_k) - \nabla F(v_k)$ so that we can write $g_t = \nabla f(w_t) - \nabla f(v_k) + \nabla F(v_k) + b_t$ instead of $g_t = \nabla f(w_t) - \nabla f(v_k) + \nabla\hat{F}(v_k)$. Factoring out the term $b_t$ and writing $g_t$ in this way makes clearer how we are able to apply the analysis of biased gradients to the analysis of online learning in the previous section. We analyze Algorithm 2 for online learning algorithms $\mathcal{A}$ that obtain *second-order* regret guarantees, $R_T(w_\star) \le \psi(w_\star)\sqrt{\sum_{t=1}^{T}\|g_t\|^2}$ as well as ones that obtain *first-order* regret guarantees $R_T(w_\star) \le \psi(w_\star)\sqrt{\sum_{t=1}^{T}\|g_t\|}$. Algorithm in these families include the well-known AdaGrad [8] algorithm and its unconstrained variant SOLO [15] (second order, $\psi(w_\star) = O(\|w_\star\|^2)$) as well as FreeRex [6] and PiSTOL [14] (first order, $\psi(w_\star) = \tilde{O}(\|w_\star\|)$). We will show that for sufficiently small $b_t$, such algorithms result in $\overline{w} = \frac{1}{T}\sum_{t=1}^{T} w_t$ such that $\mathbb{E}[F(\overline{w}) - F(w_\star)] = O(1/T)$.

---

**Algorithm 2** Online Learning with Biased Variance-Reduced Gradients

---

> **Initialize:** initial point $w_1$, epoch lengths $0 = T_0, T_1, \ldots, T_K$ online learning algorithm $\mathcal{A}$.
> Get initial vector $w_1$ from $\mathcal{A}$.
> **for** $k = 1$ **to** $K$ **do**
>     **for** $t = T_{0:k-1} + 1$ **to** $T_{0:k}$ **do**
>         Sample $f \sim \mathcal{D}$.
>         $g_t \leftarrow \nabla f(w_t) - \nabla f(v_k) + \nabla F(v_k) + b_t$.
>         Send $g_t$ to the online learning algorithm $\mathcal{A}$.
>         Get updated vector $w_{t+1}$ from $\mathcal{A}$.
>     **end for**
>     $v_{k+1} \leftarrow \frac{1}{T_k}\sum_{t=T_{1:k-1}+1}^{T_{1:k}} w_t$.
> **end for**

---

**Proposition 5.** *If $T_k/T_{k-1} \le \rho$ for all $k$, then*

$$\sum_{k=1}^{K} T_k\,\mathbb{E}[F(v_k) - F(w_\star)] \le T_1\,\mathbb{E}[F(v_1) - F(w_\star)] + \rho\sum_{t=1}^{T} \mathbb{E}[F(w_t) - F(w_\star)]$$

*Proof.* First we show that for all $k > 1$,

$$T_k\,\mathbb{E}[F(v_k) - F(w_\star)] \le \rho\sum_{t=T_{0:k-2}+1}^{T_{0:k-1}} \mathbb{E}[F(w_t) - F(w_\star)]$$

This follows from Jensen's inequality by convexity of $F$:

$$\mathbb{E}[F(v_k) - F(w_\star)] = \mathbb{E}\left[F\left(\frac{1}{T_{k-1}}\sum_{t=T_{0:k-2}+1}^{T_{0:k-1}} w_t\right) - F(w_\star)\right]$$

$$\leq \frac{1}{T_{k-1}}\mathbb{E}\left[\sum_{t=T_{0:k-2}+1}^{T_{0:k-1}} F(w_t) - \mathbb{E}[F(w_\star)]\right]$$

$$\leq \frac{\rho}{T_k}\sum_{t=T_{0:k-2}+1}^{T_{0:k-1}} \mathbb{E}[F(w_t) - F(w_\star)]$$

Now the result of the proposition is immediate:

$$\sum_{k=1}^{K} T_k\,\mathbb{E}[F(v_k) - F(w_\star)] \leq T_1\,\mathbb{E}[F(v_1) - F(w_\star)] + \sum_{k=2}^{K}\rho\sum_{t=T_{0:k-2}}^{T_{0:k-1}} \mathbb{E}[F(w_t) - F(w_\star)]$$

$$\leq T_1\,\mathbb{E}[F(v_1) - F(w_\star)] + \sum_{k=0}^{K}\rho\sum_{t=T_{0:k}}^{T_{0:k}} \mathbb{E}[F(w_t) - F(w_\star)]$$

$$= T_1\,\mathbb{E}[F(v_1) - F(w_\star)] + \rho\sum_{t=1}^{T} \mathbb{E}[F(w_t) - F(w_\star)]$$

$\square$

**Proposition 6.** *Suppose the online learning algorithm $\mathcal{A}$ guarantees regret $R_T(w_\star) \leq \psi(w_\star)\sqrt{\sum_{t=1}^{T}\|g_t\|^2}$, and $T_k/T_{k-1} \leq \rho$ for some constant $\rho$. Then*

$$\mathbb{E}\left[\sum_{t=1}^{T} F(w_t) - F(w_\star)\right] \leq \psi(w_\star)\sqrt{8(1+\rho)L\sum_{t=1}^{T}\mathbb{E}[F(w_t) - F(w_\star)] + 8LT_1\,\mathbb{E}[F(v_1) - F(w_\star)] + 2\sum_{t=1}^{T}\mathbb{E}[\|b_t\|^2]}$$

$$+ \sum_{t=1}^{T}\mathbb{E}[\|b_t\|(\|w_t - w_\star\|)]$$

*Proof.* The proof follows by applying, in order, Propositions 1, 4, and 5:

$$\mathbb{E}[\sum_{t=1}^{T} F(w_t) - F(w_\star)] \leq \mathbb{E}[R_T(w_\star)] + \sum_{t=1}^{T}\mathbb{E}[\|b_t\|(\|w_t - w_\star\|)]$$

$$\leq \psi(w_\star)\sqrt{\sum_{t=1}^{T}\|g_t\|^2} + \sum_{t=1}^{T}\mathbb{E}[\|b_t\|(\|w_t - w_\star\|)]$$

$$\leq \psi(w_\star)\sqrt{8L\sum_{t=1}^{T}\mathbb{E}[F(w_t) - F(w_\star)] + 8L\sum_{k=1}^{K}T_k\,\mathbb{E}[F(v_k) - F(w_\star)] + 2\sum_{t=1}^{T}\mathbb{E}[\|b_t\|^2]}$$

$$+ \sum_{t=1}^{T}\mathbb{E}[\|b_t\|(\|w_t - w_\star\|)]$$

$$\leq \psi(w_\star)\sqrt{8(1+\rho)L\sum_{t=1}^{T}\mathbb{E}[F(w_t) - F(w_\star)] + 8LT_1\,\mathbb{E}[F(v_1) - F(w_\star)] + 2\sum_{t=1}^{T}\mathbb{E}[\|b_t\|^2]}$$

$$+ \sum_{t=1}^{T}\mathbb{E}[\|b_t\|(\|w_t - w_\star\|)]$$

$\square$

In order to use the above Proposition 6, we need a small technical result:

**Proposition 7.** *If $a$, $b$, $c$ and $d$ are non-negative constants such that*

$$x \le a\sqrt{bx + c} + d$$

*Then*

$$x \le 4a^2 b + 2a\sqrt{c} + 2d$$

*Proof.* Suppose $x \ge 2d$. Then we have

$$\frac{x}{2} \le x - d \le a\sqrt{bx + c}$$
$$x^2 \le 4a^2 bx + 4a^2 c$$

Now we use the quadratic formula to obtain

$$x \le \frac{4a^2 b}{2} + \frac{\sqrt{16a^4 b^2 + 16a^2 c}}{2}$$
$$\le 4a^2 b + 2a\sqrt{c}$$

Since we assumed $x \ge 2d$ to obtain this bound, we conclude that $x$ is at most the maximum of $4a^2 b + 2a\sqrt{c}$ and $2d$, which is bounded by their sum. $\qquad\square$

Now we apply this result to obtain the formal statement of Proposition 2:

**Proposition 8.** *Suppose the online learning algorithm $\mathcal{A}$ guarantees regret $R_T(w_\star) \le \psi(w_\star)\sqrt{\sum_{t=1}^{T}\|g_t\|^2}$. Then for $\overline{w} = \frac{1}{T}\sum_{t=1}^{T} w_t$,*

$$\mathbb{E}[F(\overline{w}) - F(w_\star)] \le \frac{32(1+\rho)\psi(w_\star)^2 L}{T} + \frac{2\psi(w_\star)\sqrt{8LT_1\,\mathbb{E}[F(v_1) - F(w_\star)] + 2\sum_{t=1}^{T}\mathbb{E}[\|b_t\|^2]}}{T}$$
$$+ \frac{2\sum_{t=1}^{T}\mathbb{E}[\|b_t\|(\|w_t - w_\star\|)]}{T}$$

*In particular, if $\|b_t\| \le \frac{\sigma}{T}$ for all $t$ for some $\sigma$, and $V$ has diameter $D$, then*

$$\mathbb{E}[F(\overline{w}) - F(w_\star)] \le \frac{32(1+\rho)\psi(w_\star)^2 L}{T} + \frac{4\psi(w_\star)\sigma}{T\sqrt{T}} + \frac{8\psi(x)\sqrt{T_1\,\mathbb{E}[F(v_1) - F(w_\star)]}}{T} + 2\frac{\sigma D}{T}$$

*Proof.* Applying Propositions 5 to the result of Proposition 6, we have

$$\mathbb{E}\left[\sum_{t=1}^{T} F(w_t) - F(w_\star)\right] \le 32(1+\rho)\psi(w_\star)^2 L + 2\psi(w_\star)\sqrt{8LT_1\,\mathbb{E}[F(v_1) - F(w_\star)] + 2\sum_{t=1}^{T}\mathbb{E}[\|b_t\|^2]}$$
$$+ 2\sum_{t=1}^{T}\mathbb{E}[\|b_t\|(\|w_t - w_\star\|)]$$

Now divide by $T$ and use Jensen's inequality to conclude the first result. The second follows since $\sqrt{a+b} \le \sqrt{2}(\sqrt{a} + \sqrt{b})$ for all positive $a, b$. $\qquad\square$

Now let's prove bounds using the regularization trick from Proposition 3 that allows us to avoid assuming a finite-diameter domain.

**Proposition 9.** *Suppose the online learning algorithm $\mathcal{A}$ guarantees regret $R_T(w_\star) \le \psi(w_\star)\sqrt{\sum_{t=1}^{T}\|g_t\|^2}$. Let $B$ be a uniform upper bound on $\|b_t\|$. Run $\mathcal{A}$ on the gradients $\|g_t\| + B\frac{w_t}{\|w_t\|}$. Then for $\overline{w} = \frac{1}{T}\sum_{t=1}^{T} w_t$,*

$$\mathbb{E}[F(\overline{w}) - F(w_\star)] \le \frac{64(1+\rho)\psi(w_\star)^2 L}{T} + \frac{2\psi(w_\star)\sqrt{16LT_1\,\mathbb{E}[F(v_1) - F(w_\star)] + 6TB^2}}{T} + 2B\|w_\star\|$$

*In particular, if $B \le \frac{\sigma}{T}$, we have*

$$\mathbb{E}[F(\overline{w}) - F(w_\star)] \le \frac{64(1+\rho)\psi(w_\star)^2 L}{T} + \frac{2\psi(w_\star)\sqrt{16LT_1\,\mathbb{E}[F(v_1) - F(w_\star)] + 6\sigma/T}}{T} + 2\frac{\sigma\|w_\star\|}{T}$$

*Proof.*

$$\mathbb{E}[\sum_{t=1}^{T} F(w_t) - F(w_\star)] \leq \mathbb{E}[R_T(w_\star)] + \sum_{t=1}^{T} \mathbb{E}[b_t \cdot (w_\star - w_t)] + B\|w_\star\| - B\|w_t\|$$

$$\leq \mathbb{E}[R_T(w_\star)] + 2TB\|w_\star\|$$

$$\leq \psi(w_\star)\sqrt{\sum_{t=1}^{T} 2\,\mathbb{E}[\|g_t\|^2] + 2TB^2} + 2TB\|w_\star\|$$

$$\leq \psi(w_\star)\sqrt{16L\sum_{t=1}^{T}\mathbb{E}[F(w_t) - F(w_\star)] + 16L\sum_{k=1}^{K} T_k\,\mathbb{E}[F(v_k) - F(w_\star)] + 4\sum_{t=1}^{T}\mathbb{E}[\|b_t\|^2] + 2TB^2}$$
$$+ 2TB\|w_\star\|$$

$$\leq \psi(w_\star)\sqrt{16(1+\rho)L\sum_{t=1}^{T}\mathbb{E}[F(w_t) - F(w_\star)] + 16LT_1\,\mathbb{E}[F(v_1) - F(w_\star)] + 4\sum_{t=1}^{T}\mathbb{E}[\|b_t\|^2] + 2TB^2}$$
$$+ 2TB\|w_\star\|$$

$$\leq \psi(w_\star)\sqrt{16(1+\rho)L\sum_{t=1}^{T}\mathbb{E}[F(w_t) - F(w_\star)] + 16LT_1\,\mathbb{E}[F(v_1) - F(w_\star)] + 6TB^2}$$
$$+ 2TB\|w_\star\|$$

Now use Proposition 7:

$$\mathbb{E}[\sum_{t=1}^{T} F(w_t) - F(w_\star)] \leq 64(1+\rho)\psi(w_\star)^2 L + 2\psi(w_\star)\sqrt{16LT_1\,\mathbb{E}[F(v_1) - F(w_\star)] + 6TB^2} + 2TB\|w_\star\|$$

$\square$

We can use Proposition 9 to prove an analogue of Theorem 1:

**Theorem 3.** *Suppose the online learning algorithm $\mathcal{A}$ guarantees regret $R_T(w_\star) \leq \psi(w_\star)\sqrt{\sum_{t=1}^{T}\|g_t\|^2}$.*
*Set $b_t = \|\nabla\hat{F}(v_k) - \nabla F(v_k)\|$ for $t \in [T_{0:k-1}+1, T_{1:k}]$. Suppose that $T_k/T_{k-1} \leq \rho$ for all $k$. Let $B$ be a uniform upper bound on $\|b_t\|$. Run $\mathcal{A}$ on the gradients $g_t + B\frac{w_t}{\|w_t\|}$. Then for $\overline{w} = \frac{1}{T}\sum_{t=1}^{T} w_t$,*

$$\mathbb{E}[F(\overline{w}) - F(w_\star)] \leq 2B\|w_\star\| + \frac{64(1+\rho)\psi(w_\star)^2 L}{T} + \frac{2\psi(w_\star)\sqrt{16LT_1\,\mathbb{E}[F(v_1)-F(w_\star)]+6TB^2}}{T}$$

# E   Other Types of Regret Bounds

The above Theorems 1 and 3 postulate a regret bound of the form $R_T(w_\star) \leq \psi(w_\star)\sqrt{\sum_{t=1}^{T}\|g_t\|^2}$. This bound is achieved by, e.g. the AdaGrad [8] or SOLO [15] algorithms, both of which have $\psi(w_\star) = O(\|w_\star\|^2)$. However, there also exist algorithms (e.g. PiSTOL [14] or FreeRex [6]) that improve $\psi$ to $O(\|x\|)$ up to log factors, but in return achieve only a *first-order* regret bound of $R_T(w_\star) \leq \psi(w_\star)\sqrt{G\sum_{t=1}^{T}\|g_t\|}$, where $G$ is a uniform bound on $\|g_t\|$. Such algorithms are sometimes called parameter-free algorithms because they achieve the minimax optimal regret bound of $\|w_\star\|G\sqrt{T}$ in an unconstrained setting without requiring any hyperparameter tuning. In this section we show similar convergence guarantees for these parameter-free algorithms.

We start with a simple LaGrange multipliers argument:

**Proposition 10.** *Suppose $a_1, \dots, a_t$ are non-negative real numbers. Then*

$$\sum_{t=1}^{T} a_t \leq \sqrt{T}\sqrt{\sum_{t=1}^{T} a_t^2}$$

*Proof.* Let $S = \sum_{t=1}^{T} a_t^2$. We want to maximize $\sum_{t=1}^{T} a_t$ subject to $\sum_{t=1}^{T} a_t^2 \leq S$. Then applying the method of LaGrange multipliers, we see that for the optimizing values of $a_t$, there is some multiplier $\lambda$ such that for all $t$, either $\lambda = 2a_t$ or $a_t = 0$. Therefore $a_t \leq \sqrt{S/T}$ and so $\sum_{t=1}^{T} a_t \leq \sqrt{TS} = \sqrt{T}\sqrt{\sum_{t=1}^{T} a_t^2}$. $\square$

Using Proposition 10, we can prove an analogue of Proposition 6:

**Proposition 11.** *Suppose the online learning algorithm $\mathcal{A}$ guarantees regret $R_T(w_\star) \leq \psi(w_\star)\sqrt{\sum_{t=1}^{T} \|g_t\|}$, and $T_k/T_{k-1} \leq \rho$ for some constant $\rho$. Then*

$$\mathbb{E}\left[\sum_{t=1}^{T} F(w_t) - F(w_\star)\right] \leq \psi(w_\star)\sqrt{\sqrt{T}\sqrt{8(1+\rho)L\sum_{t=1}^{T}\mathbb{E}[F(w_t) - F(w_\star)] + 8LT_1\,\mathbb{E}[F(v_1) - F(w_\star)] + 2\sum_{t=1}^{T}\mathbb{E}[\|b_t\|^2]}}$$

$$+ \sum_{t=1}^{T} \mathbb{E}[\|b_t\|(\|w_t - w_\star\|)]$$

*Proof.* The proof is nearly identical to that of Proposition 6:

$$\mathbb{E}[\sum_{t=1}^{T} F(w_t) - F(w_\star)] \leq \mathbb{E}[R_T(w_\star)] + \sum_{t=1}^{T}\mathbb{E}[\|b_t\|(\|w_t - w_\star\|)]$$

$$\leq \psi(w_\star)\sqrt{\sum_{t=1}^{T}\|g_t\|} + \sum_{t=1}^{T}\mathbb{E}[\|b_t\|(\|w_t - w_\star\|)]$$

$$\leq \psi(w_\star)\sqrt{\sqrt{T}\sqrt{\sum_{t=1}^{T}\|g_t\|^2}} + \sum_{t=1}^{T}\mathbb{E}[\|b_t\|(\|w_t - w_\star\|)]$$

$$\leq \psi(w_\star)\sqrt{\sqrt{T}\sqrt{8L\sum_{t=1}^{T}\mathbb{E}[F(w_t) - F(w_\star)] + 8L\sum_{k=1}^{K}T_k\,\mathbb{E}[F(v_k) - F(w_\star)] + 2\sum_{t=1}^{T}\mathbb{E}[\|b_t\|^2]}}$$

$$+ \sum_{t=1}^{T}\mathbb{E}[\|b_t\|(\|w_t - w_\star\|)]$$

$$\leq \psi(w_\star)\sqrt{\sqrt{T}\sqrt{8(1+\rho)L\sum_{t=1}^{T}\mathbb{E}[F(w_t) - F(w_\star)] + 8LT_1\,\mathbb{E}[F(v_1) - F(w_\star)] + 2\sum_{t=1}^{T}\mathbb{E}[\|b_t\|^2]}}$$

$$+ \sum_{t=1}^{T}\mathbb{E}[\|b_t\|(\|w_t - w_\star\|)]$$

$\square$

Now in order to use Proposition 11 we need a slightly different technical result than Proposition 7:

**Proposition 12.** *If a, b, c, d and e, and f are non-negative constants such that*

$$x \leq a\sqrt{b\sqrt{cx + d} + e} + f$$

*then*

$$x \leq 2^{7/3}a^{4/3}b^{2/3}c^{1/3} + 2a\sqrt{2b\sqrt{2d} + 2e} + 2f$$

*Proof.* Since $\sqrt{x + y} \leq \sqrt{2x} + \sqrt{2y}$ for all non-negative $x$ and $y$, we have

$$x \leq a\sqrt{b\sqrt{2cx} + b\sqrt{2d} + e} + f$$

$$\leq a\sqrt{2b\sqrt{2cx}} + a\sqrt{2b\sqrt{2d} + 2e} + f$$

Now suppose $x \geq 2a\sqrt{2b\sqrt{2d} + 2e} + 2f$. Then

$$\frac{x}{2} \leq a\sqrt{2b\sqrt{2cx}}$$

$$x^4 \leq 2^7 a^4 b^2 cx$$

$$x \leq 2^{7/3}a^{4/3}b^{2/3}c^{1/3}$$

So that taking into account our condition $x \geq 2a\sqrt{2b\sqrt{2d} + 2e} + 2f$, we have

$$x \leq 2^{7/3}a^{4/3}b^{2/3}c^{1/3} + 2a\sqrt{2b\sqrt{2d} + 2e} + 2f$$

as desired. $\qquad\square$

With this in hand, we can prove an analogue of Proposition 8:

**Proposition 13.** *Suppose the online learning algorithm $\mathcal{A}$ guarantees regret $R_T(w_\star) \leq \psi(w_\star)\sqrt{\sum_{t=1}^T \|g_t\|}$. Then for $\overline{w} = \frac{1}{T}\sum_{t=1}^T w_t$,*

$$\mathbb{E}[F(\overline{w}) - F(w_\star)] \leq \frac{2^{10/3}\psi(x^*)^{4/3}(1+\rho)^{1/3}L^{1/3}}{T^{2/3}}$$

$$+ \frac{2^{7/4}\psi(x^*)\sqrt{8LT_1\,\mathbb{E}[F(v_1) - F(w_\star)] + 2\sum_{t=1}^T \mathbb{E}[\|b_t\|^2]}}{T^{3/4}} + 2\frac{\sum_{t=1}^T \mathbb{E}[\|b_t\|(\|w_t - w_\star\|)]}{T}$$

*In particular, if $\|b_t\| \leq \frac{\sigma}{T^{2/3}}$ for all t for some $\sigma$, and $V$ has diameter $D$, then*

$$\mathbb{E}[F(\overline{w}) - F(w_\star)] \leq \frac{2^{10/3}\psi(x^*)^{4/3}(1+\rho)^{1/3}L^{1/3}}{T^{2/3}}$$

$$+ \frac{2^{11/4}\sigma}{T^{13/12}} + \frac{2^{15/4}\psi(x^*)\sqrt{LT_1\,\mathbb{E}[F(v_1) - F(w_\star)]}}{T^{3/4}} + 2\frac{\sigma D}{T}$$

*Proof.* The proof is directly analogous to the proof of Proposition 8; we simply apply Proposition 12 to Proposition 11:

$$\mathbb{E}\left[\sum_{t=1}^T F(w_t) - F(w_\star)\right] \leq \psi(w_\star)\sqrt{\sqrt{T}\sqrt{8(1+\rho)L\sum_{t=1}^T \mathbb{E}[F(w_t) - F(w_\star)] + 8LT_1\,\mathbb{E}[F(w_1) - F(w_\star)] + 2\sum_{t=1}^T \mathbb{E}[\|b_t\|^2]}}$$

$$+ \sum_{t=1}^T \mathbb{E}[\|b_t\|(\|w_t - w_\star\|)]$$

$$\mathbb{E}\left[\sum_{t=1}^T F(w_t) - F(w_\star)\right] \leq 2^{10/3}\psi(x^*)^{4/3}T^{1/3}(1+\rho)^{1/3}L^{1/3}$$

$$+ 2^{7/4}\psi(x^*)T^{1/4}\sqrt{8LT_1\,\mathbb{E}[F(w_1) - F(w_\star)] + 2\sum_{t=1}^T \mathbb{E}[\|b_t\|^2] + 2\sum_{t=1}^T \mathbb{E}[\|b_t\|(\|w_t - w_\star\|)]}$$

Now divide by $T$ to obtain the result. $\qquad\square$

Finally, we prove the analogue of Proposition 9:

**Proposition 14.** *Suppose the online learning algorithm $\mathcal{A}$ guarantees regret $R_T(w_\star) \leq \psi(w_\star)\sqrt{\sum_{t=1}^T \|g_t\|^2}$. Let $B$ be a uniform upper bound on $\|b_t\|$. Run $\mathcal{A}$ on the gradients $\|g_t\| + B\frac{w_t}{\|w_t\|}$. Then for $\overline{w} = \frac{1}{T}\sum_{t=1}^T w_t$,*

$$\mathbb{E}[F(\overline{w}) - F(w_\star)] \leq 2^4\psi(x^*)^{4/3}T^{1/3}(1+\rho)^{1/3}L^{1/3}$$

$$+ 2^{9/4}\psi(x^*)T^{1/4}\sqrt{8LT_1\,\mathbb{E}[F(v_1) - F(w_\star)] + 2TB^2} + 2\psi(w_\star)\sqrt{2TB} + 4TB\|w_\star\|$$

*In particular, if $B \leq \frac{\sigma}{T^{2/3}}$, we have*

$$\mathbb{E}[F(\overline{w}) - F(w_\star)] \leq \frac{2^4\psi(x^*)^{4/3}(1+\rho)^{1/3}L^{1/3}}{T^{2/3}}$$

$$+ \frac{2^{17/4}\psi(x^*)\sqrt{LT_1\,\mathbb{E}[F(v_1) - F(w_\star)]}}{T^{3/4}} + \frac{2^{13/4}\psi(x^*)\sigma}{T^{11/12}} + \frac{2\psi(w_\star)\sqrt{2\sigma}}{T^{5/6}} + \frac{4\|w_\star\|}{T^{2/3}}$$

*Proof.*

$$\mathbb{E}[\sum_{t=1}^{T} F(w_t) - F(w_\star)] \leq \mathbb{E}[R_T(w_\star)] + \sum_{t=1}^{T} \mathbb{E}[b_t \cdot (w_\star - w_t)] + B\|w_\star\| - B\|w_t\|$$

$$\leq \mathbb{E}[R_T(w_\star)] + 2TB\|w_\star\|$$

$$\leq \psi(w_\star)\sqrt{\sum_{t=1}^{T} \mathbb{E}[\|g_t\|] + TB} + 2TB\|w_\star\|$$

$$\leq \psi(w_\star)\sqrt{2\sum_{t=1}^{T} \mathbb{E}[\|g_t\|]} + \psi(w_\star)\sqrt{2TB} + 2TB\|w_\star\|$$

$$\leq \psi(w_\star)\sqrt{2\sqrt{T}\sqrt{8L\sum_{t=1}^{T} \mathbb{E}[F(w_t) - F(w_\star)] + 8L\sum_{k=1}^{K} T_k\,\mathbb{E}[F(v_k) - F(w_\star)] + 2\sum_{t=1}^{T} \mathbb{E}[\|b_t\|^2]}}$$
$$+ \psi(w_\star)\sqrt{2TB} + 2TB\|w_\star\|$$

$$\leq \psi(w_\star)\sqrt{2\sqrt{T}\sqrt{8(1+\rho)L\sum_{t=1}^{T} \mathbb{E}[F(w_t) - F(w_\star)] + 8LT_1\,\mathbb{E}[F(v_1) - F(w_\star)] + 2\sum_{t=1}^{T} \mathbb{E}[\|b_t\|^2]}}$$
$$+ \psi(w_\star)\sqrt{2TB} + 2TB\|w_\star\|$$

Now we apply Proposition 12:

$$\mathbb{E}[\sum_{t=1}^{T} F(w_t) - F(w_\star)] \leq 2^4 \psi(x^*)^{4/3} T^{1/3}(1+\rho)^{1/3} L^{1/3}$$

$$+ 2^{9/4}\psi(x^*)T^{1/4}\sqrt{8LT_1\,\mathbb{E}[F(v_1) - F(w_\star)] + 2\sum_{t=1}^{T} \mathbb{E}[\|b_t\|^2] + 2\psi(w_\star)\sqrt{2TB} + 4TB\|w_\star\|}$$

$$\leq 2^4 \psi(x^*)^{4/3} T^{1/3}(1+\rho)^{1/3} L^{1/3}$$
$$+ 2^{9/4}\psi(x^*)T^{1/4}\sqrt{8LT_1\,\mathbb{E}[F(v_1) - F(w_\star)] + 2TB^2} + 2\psi(w_\star)\sqrt{2TB} + 4TB\|w_\star\|$$

$$\square$$

Combining all these together, we can prove analogues of Theorems 1 and 3:

**Theorem 4.** *Suppose the online learning algorithm $\mathcal{A}$ guarantees regret $R_T(w_\star) \leq \psi(w_\star)\sqrt{\sum_{t=1}^{T} \|g_t\|}$.*
*Set $b_t = \|\nabla\hat{F}(v_k) - \nabla F(v_k)\|$ for $t \in [T_{0:k-1}+1, T_{1:k}]$. Suppose that $T_k/T_{k-1} \leq \rho$ for all $k$. Then for $\overline{w} = \frac{1}{T}\sum_{t=1}^{T} w_t$,*

$$\mathbb{E}[F(\overline{w}) - F(w_\star)] \leq \frac{2^{10/3}\psi(x^*)^{4/3}(1+\rho)^{1/3}L^{1/3}}{T^{2/3}} + \frac{2^{7/4}\psi(x^*)\sqrt{8LT_1\,\mathbb{E}[F(v_1) - F(w_\star)] + 2\sum_{t=1}^{T} \mathbb{E}[\|b_t\|^2]}}{T^{3/4}}$$
$$+ 2\frac{\sum_{t=1}^{T} \mathbb{E}[\|b_t\|(\|w_t - w_\star\|)]}{T}$$

*In particular, with probability at least $1 - \frac{1}{T}$ we have $\|b_t\| \leq \frac{\sigma\sqrt{\log(KT)}}{\sqrt{\hat{N}}}$ for some $\sigma$, and if $\hat{N} > T^{4/3}$ and $V$ has diameter $D$, then*

$$\mathbb{E}[F(\overline{w}) - F(w_\star)] \leq \frac{2^{10/3}\psi(x^*)^{4/3}(1+\rho)^{1/3}L^{1/3}}{T^{2/3}} + \frac{2^{11/4}\sigma\sqrt{\log(KT)}}{T^{13/12}}$$
$$+ \frac{2^{15/4}\psi(x^*)\sqrt{LT_1\,\mathbb{E}[F(v_1) - F(w_\star)]}}{T^{3/4}} + 2\frac{\sigma\sqrt{\log(KT)}D}{T} + \frac{GD}{T}$$
$$= O\left(\frac{\sqrt{\log(KT)}}{T} + \frac{1}{T^{2/3}}\right)$$

**Theorem 5.** *Suppose the online learning algorithm $\mathcal{A}$ guarantees regret $R_T(w_\star) \leq \psi(w_\star)\sqrt{\sum_{t=1}^T \|g_t\|}$. Set $b_t = \|\nabla \hat{F}(v_k) - \nabla F(v_k)\|$ for $t \in [T_{0:k-1} + 1, T_{1:k}]$. Suppose that $T_k/T_{k-1} \leq \rho$ for all $k$. Let $B$ be a uniform upper bound on $\|b_t\|$. Run $\mathcal{A}$ on the gradients $g_t + B\frac{w_t}{\|w_t\|}$. Then for $\overline{w} = \frac{1}{T}\sum_{t=1}^T w_t$,*

$$\mathbb{E}[F(\overline{w}) - F(w_\star)] \leq \frac{2^4 \psi(x^*)^{4/3}(1+\rho)^{1/3}L^{1/3}}{T^{2/3}} + \frac{2^{9/4}\psi(x^*)\sqrt{8LT_1\,\mathbb{E}[F(v_1) - F(w_\star)] + 2TB^2}}{T^{3/4}}$$

$$+ \frac{2\psi(w_\star)\sqrt{2B}}{\sqrt{T}} + 4B\|w_\star\|$$

We also have an analogue of Theorem 2 using essentially the same argument, but using the setting $\hat{N} = \Theta(T^{4/3})$ rather than $\hat{N} = \Theta(T^2)$:

**Theorem 6.** *Set $T_k = 2T_{k-1}$. Suppose the base optimizer $\mathcal{A}$ in SVRG OL guarantees regret $R_T(w_\star) \leq \psi(w_\star)\sqrt{\sum_{t=1}^T \|g_t\|}$, and the domain $W$ has finite diameter $D$. Let $\hat{N} = \Theta(T^{4/3})$ and $N = K\hat{N} + T$ be the total number of data points observed. Suppose we compute the batch gradients $\nabla \hat{F}(v_k)$ in parallel on $m$ machines with $m < N^{1/4}$. Then for $\overline{w} = \frac{1}{T}\sum_{t=1}^T w_t$ we obtain*

$$\mathbb{E}[F(\overline{w}) - F(w_\star)] = \tilde{O}\left(\frac{1}{\sqrt{N}}\right)$$

*in time $\tilde{O}(N/m)$ and space $O(1)$*

## F  Benchmark results

Table 4: Average loss and AUC achieved by Logistic regression implemented in Spark ML, VW and SVRG OL. "Comm." refers to number of communication rounds and time is measured in minutes. In case of MiniBatch SGD and SVRG, we computed the full batch gradient in each iteration based on the whole training data.

| Data | Comm. | Train | Test | AUC | Time |
|---|---|---|---|---|---|
| | | | KDD10 | | |
| Spark ML | 100 | 0.25380 | 0.26317 | 84.778 | 52 |
| Spark ML | 500 | 0.25303 | **0.26234** | **84.880** | 101 |
| VW | 100 | 0.24462 | 0.26849 | 84.525 | 38 |
| VW | 500 | **0.18879** | 0.27817 | 83.967 | 96 |
| MiniBatch SGD | 100 | 0.36051 | 0.36711 | 80.603 | 33 |
| MiniBatch SGD | 500 | 0.35292 | 0.359483 | 80.932 | 121 |
| SVRG | 100 | 0.31743 | 0.32981 | 81.121 | 35 |
| SVRG | 500 | 0.31354 | 0.32346 | 81.928 | 102 |
| SVRG OL | 4 | 0.26085 | 0.26525 | 84.459 | 6 |
| | | | KDD12 | | |
| Spark ML | 100 | 0.15756 | 0.15589 | 75.485 | 36 |
| Spark ML | 500 | 0.15755 | 0.15570 | 75.453 | 180 |
| VW | 100 | 0.15398 | 0.15725 | 77.871 | 44 |
| VW | 500 | **0.14866** | 0.15550 | **78.881** | 150 |
| MiniBatch SGD | 100 | 0.27689 | 0.27420 | 70.207 | 43 |
| MiniBatch SGD | 500 | 0.27420 | 0.27689 | 70.207 | 225 |
| SVRG | 100 | 0.24325 | 0.23754 | 72.543 | 54 |
| SVRG | 500 | 0.22784 | 0.22397 | 73.764 | 202 |
| SVRG OL | 4 | 0.152740 | **0.154985** | 78.431 | 8 |
| | | | ADS SMALL | | |
| Spark ML | 100 | 0.23372 | 0.22288 | 83.356 | 42 |
| Spark ML | 500 | 0.23365 | 0.22286 | 83.365 | 245 |
| VW | 100 | 0.23381 | 0.22347 | 83.214 | 114 |
| VW | 500 | 0.23157 | 0.22251 | **83.499** | 396 |
| SVRG OL | 14 | **0.23147** | **0.22244** | 83.479 | 94 |
| Data | | | ADS LARGE | | |
| Spark ML | 100 | 0.23965 | 0.23263 | 82.646 | 216 |
| Spark ML | 500 | 0.23958 | 0.23256 | 82.655 | 1214 |
| VW | 100 | | | | FAIL |
| SVRG OL | 31 | **0.23753** | **0.23197** | **82.830** | 334 |
| Data | | | EMAIL | | |
| Spark ML | 100 | 0.27837 | 0.29598 | 88.852 | 117 |
| Spark ML | 500 | **0.27773** | **0.28887** | **88.863** | 601 |
| VW | 100 | 0.35812 | 0.33919 | 84.414 | 74 |
| VW | 500 | 0.33094 | 0.33010 | 85.854 | 358 |
| SVRG OL | 14 | 0.30567 | 0.29889 | 88.321 | 110 |