[Reviews · NeurIPS 2018]

Reviewer 1



Update: I keep my initial rating. As a potential improvement for the paper, I see the authors only tackle the non strictly convex case. I am curious how the result would be modified if the authors assumed strong convexity. Original review: In this paper the authors introduce SVRG OL, a distributed stochastic optimization method for convex optimization. Inspired by SVRG, the authors first compute a high precision estimate of a gradient at an anchor point v using a large number of samples. This part of the computation can be trivially distributed as a map reduce job. Then, the authors perform a relatively small number of serial adaptative SGD updates (i.e. adagrad) on a single machine but with modified gradients similarly to what is done in SVRG but with the high precision estimate of the full gradient at point v. They then use the average of the iterates of the adaptative SGD as a new starting point and anchor point v. For a given budget N, roughly sqrt(N) iterations of adaptative SGD are performed. The computation of the gradient is requires close to N computation of gradients which are dispatched on m machines. As long as m < sqrt(N), the SGD part is not limitant thus allowing for a linear speedup when increasing m. I am wondering what is the critical contribution of this paper. In particular the number of samples used to compute the full gradient estimate seems to be of the order N, the dataset size, so there seem to be a limited gain on that side. The other aspect is the adaptative SGD inner iterations. But again I did not get a feeling of how this method would compare to simply using SVRG with a distributed computation of the full gradient and then serial updates. My understanding is that SVRG OL is some interpolation between SGD and SVRG, carefully done to obtain a final convergence rate of 1/sqrt(N). I also have some doubt about practical applications. My own experience with SVRG was that it was not significantly outperforming SGD for complex task and it seems that this algorithm uses a very few number of samples for the SGD iterations. My intuition is that typically in deep learning, small batch sizes works better and in order to learn efficiently from a training example, one should ideally compute its gradient individually. In the present algorithm, the vast majority of the data points will only be used to reduce the variance of other data points gradients but will never themselves drive an update. I'm interested in any comment from the authors on this surprising affect. The experimental part is using large datasets, which is a good thing for distributed optimization. There seem to be a gap between the theory and practice. The results looks promising but the presentation as a table is not so easy to read, it seems that the different algorithms are run for different number of epochs making comparison impossible in the table. For instance if algorithm A as run for twice less time and has a worst accuracy then model B that has run for twice as long as A, then no conclusion can be drawn. In my opinion a table is only useful if the same budget is given for all methods for instance: - accuracy after same run time, - accuracy after same number of training examples, - time to reach fixed accuracy etc. I would have enjoyed a plot of the time to reach a given accuracy, plotting this for a varying number of workers is also a very good way to verify the linear speed up. Overall I found the paper interesting. I did not manage to get a good understanding of why this approach was working while the number of SGD updates seems surprisingly small, I tried to understand the proofs techniques for that but did not have enough time to really dive into them. I am in favor of acceptance A few remarks: - in the proof of Lemma 1, Prob(||...|| >= epsilon for all k), shouldn't it be Prob(there exist k such as ||...|| > epsilon), as there exist would be the union while for all is the intersection of the probabilistic events? - in proof of Lemma 1, looking around it did not seem that a vector space version of Hoeffding was so easy to find. If the authors would have some reference for it, it would be nice.

Reviewer 2



=========== after rebuttal I carefully read the authors’ response. Some of my criticism has been addressed, yet I think that some critical flaws are still in the paper and I am not confident that these would be fixable in a final version within this submission round. I believe that the authors do not make a strong case for their algorithm, since they do not appropriately compare with the state of the art. Eg, they do not compare to any of the algorithms mentioned below: https://arxiv.org/pdf/1511.01942.pdf and https://arxiv.org/pdf/1603.06861.pdf The authors mention in their experimental section: “Implementing our algorithm in the Spark environment is fairly straightforward. SVRG OL switches between two phases: a batch gradient computation phase and a serial SGD phase. “ As I mentioned in my review, this is very very similar -modulo the varying stepsize- to the algorithms above. At the very least the authors should have compared with the above. Moreover, the authors *do not* support a key claim of the paper, ie that the proposed algorithm achieves a linear speedup? (this claim is made in the 4th sentence of their abstract, and repeated through out their text) There is no single experiment that compares the serial implementation of the algorithm with its distributed version, hence no scaling/speedup result can be inferred. Since they addressed some of my criticisms, but not the major points above, I will update my score from 3 to 4. ===============original review The authors present SVRG OL, an adaptive SVRG algorithm (not quite sure why it is refered as SGD), that is designed to get linear speedups on parallel architectures. The theoretical contribution of the paper is that the algorithm requires log number of comm rounds to achieve an epsilon accurate result. Although the problem setup is interesting, there are several issues with the paper that I list below: - First of all, the presented algorithm is *identical* to Algorithm 1 here:https://arxiv.org/pdf/1412.6606.pdf Algorithm 1 here https://arxiv.org/pdf/1511.01942.pdf and CheapSVRG here: https://arxiv.org/pdf/1603.06861.pdf It is unclear to me what is algorithmically novel between the above and the presented SVRG OL algorithm. - This is certainly subjective, but I do not understand the usefulness of online optimization results in the context of distributed setups. In most dist. setups that are of interest, all data is available for batch processing. In this case convergence rates are indeed important, but online regret bounds don’t shed a particular insight on how to implement these algorithms. - The authors mention “Unfortunately, minibatch-SGD obtains a communication complexity that scales as sqrt{N} (or N^1/4 for accelerated variants). In modern problems when N is extremely large, this overhead is prohibitively large.” This is absolutely not true, as every single large scale ML problem that trains DNNs uses backprop and minibatch SGD implementations. So although the # comm rounds is high, it is not prohibitive. - The authors mention “prior analyses require specific settings for ⌘ that incorporate L and fail to converge with incorrect settings, requiring the user to manually tune ⌘ to obtain the desired performance. “ That is true, but in practice one almost never computes L or the str. cvx constant, as that can be as expensive as solving the problem. For non-convex problems that actually *never* happens (eg pre computing L). - The authors implement the algorithm in spark. All the data sets tested fit in a single machine, and can most probably be run much faster with a simple python implementation of SVRG. I will not be convinced that the implementations presented are relevant, unless the authors have a comparison with a single machine, SVRG non-Spark baseline. - Although the authors make a claim on linear speedups, no speedup curves are presented in the paper. - minor: Although the authors mention they need log number of comm rounds, in Table 1 they present constants and omit the log factors. There is enough space to list the details in this table. Overall, this paper deals with an interesting problem. however, the presented algorithm is not novel, the implementation and experiments might not be as relevant, and comparisons with serial SVRG are missing. I do not think the paper should be accepted to NIPS.

Reviewer 3



The authors propose a distributed stochastic algorithm for expected risk minimization that achieves optimal convergence rate $N^{-1/2}$, with time $O(N/m)$ (here $m$ is the number of local machines"), space $O(1)$ and $K= O(1)$ communication rounds. The algorithm seems interesting. However, as I did not manage to read the proofs, thus I am willing to let other qualified reviewers make the final decision. Minor comments: There are much taking expectation step" in the paper. Perhaps it is better to clarify which random variable you are taking the expectation with respect to? For example, in Proposition 1, $\mathbb{E}[g_t]$ should be $\mathbb{E}[g_t | w_t]?$ In the proof of Lemma 1, $v_k$ is a random variable, and thus when applying the concentration inequalities, the event will also depend on this random variable.